# Towards Efficient Conversational Recommendations: Expected Value of Information Meets Bandit Learning

## Abstract

In conversational recommender systems, interactively presenting queries and leveraging user feedback are crucial for efficiently estimating user preferences and improving recommendation quality. Selecting optimal queries in these systems is a significant challenge that has been extensively studied as a sequential decision problem. The *expected value of information (EVOI)*, which computes the expected reward improvement, provides a principled criterion for query selection. However, it is computationally expensive and lacks theoretical performance guarantees. Conversely, *conversational bandits* offer provable regret upper bounds, but their query selection strategies yield only marginal regret improvements over non-conversational approaches. To address these limitations, we integrate EVOI within the conversational bandit framework by proposing a new conversational mechanism featuring two key techniques: (1) *gradient-based EVOI*, which replaces the complex Bayesian updates in conventional EVOI with efficient stochastic gradient descent, significantly reducing computational complexity and facilitating theoretical analysis; and (2) *smoothed key term contexts*, which enhance exploration by adding random perturbations to uncover more specific user preferences. Our approach applies to both Bayesian (Thompson Sampling) and frequentist (UCB) variants of conversational bandits. We introduce two new algorithms, `ConTS-EVOI` and `ConUCB-EVOI`, and rigorously prove that they achieve substantially tighter regret bounds, with both algorithms offering a $\sqrt{d}$ improvement in their dependence on the time horizon $T$, where $d$ is the dimension of the feature space. Extensive evaluations on synthetic and real-world datasets validate the effectiveness of our methods.

## CCS Concepts

• **Information systems** → **Recommender systems**; • **Theory of computation** → **Online learning algorithms**; **Online learning theory**.

## Keywords

Conversational Recommendation, Preference Elicitation, Multi-Armed Bandit, Online Learning

**ACM Reference Format:**
Anonymous Author(s). 2018. Towards Efficient Conversational Recommendations: Expected Value of Information Meets Bandit Learning. In *Proceedings of Make sure to enter the correct conference title from your rights confirmation emai (Conference acronym 'XX)*. ACM, New York, NY, USA, 14 pages. https://doi.org/XXXXXXX.XXXXXXX

## 1 Introduction

Recommender systems play a crucial role in personalizing content across various industries such as e-commerce, news feeds, social networks, and video streaming platforms. Over the past decade, research in this field has progressed from traditional *collaborative filtering* methods to advanced *online learning-based* approaches. By interacting with users and collecting data such as click rates and browsing histories, these systems employ sophisticated algorithms to learn user preferences and adaptively adjust their recommendations. However, one significant challenge faced by recommender systems is the *cold start* problem [5, 15], which arises when dealing with new users who lack sufficient historical data. In such cases, the system struggles to generate high-quality, personalized recommendations because it relies solely on passive user feedback, such as clicks or views, which usually take time to accumulate.

To address the cold start problem, *conversational recommender systems* [10, 31, 42] adopt a more proactive approach by initiating conversations with users. Instead of passively waiting for users to click or view, these systems actively engage users by asking questions to quickly gather information about their preferences, thus accelerating the learning process. The questions are usually related to categories of items such as movie genres and news topics, which help to rapidly narrow down the user's interests. For example, asking a direct question like "Do you enjoy comedy movies?" can provide immediate insights into a user's tastes, enabling the system to offer faster and more accurate recommendations.

Selecting optimal queries in conversational recommender systems is a critical challenge that has been widely studied as a sequential decision-making problem. A prominent line of research, known as *preference elicitation* [21], focuses on designing criteria to select queries that best elicit user preferences. One commonly used criterion is the *expected value of information (EVOI)* [16, 18, 26, 34, 35], which measures the quality of a query based on the expected improvement in recommendation quality resulting from incorporating the user's anticipated response. Intuitively, selecting queries that maximize EVOI can help elicit user preferences with minimal interactions, which is particularly valuable for reducing user fatigue and improving the overall recommendation experience. Although conceptually attractive, the EVOI method faces two major challenges. First, as a Bayesian approach, EVOI requires maintaining and updating a probabilistic model of the user's preferences based on their responses. However, performing exact posterior updates for EVOI via Bayes' rule is computationally intractable, even for simple distributions such as multivariate Gaussians. Consequently,

existing studies frequently focus on computational aspects [26, 34], and practical implementations often rely on simplifications or approximations [11, 14, 16]. Second, the typical usage of EVOI is *myopic*, meaning that it focuses on one-step planning without considering long-term outcomes. This greedy approach does not offer theoretical guarantees for long-term performance metrics such as cumulative regret, which is crucial for evaluating the overall effectiveness of sequential decision-making strategies.

Recently, another line of research, known as *conversational bandits* [10, 24, 25, 36, 40, 41], models conversational recommendation as a multi-armed bandit [22] problem. In this framework, recommendable items are modeled as "arms", and queries are modeled as conversational "key terms". Online learning strategies such as Upper Confidence Bound (UCB) [41] and Thompson Sampling (TS) [24] are employed to balance the exploration-exploitation trade-off in both query selection and item recommendation. While conversational bandits consider long-term performance and provide provable cumulative regret upper bounds, the query selection strategies proposed so far have only yielded marginal regret improvements over non-conversational bandits. For example, both the standard linear bandit algorithm LinUCB [2] and the conversational linear bandit algorithm ConUCB [41] achieve the same regret bound of order $\widetilde{O}(d\sqrt{T})$, where $d$ is the dimension of the feature space and $T$ is the time horizon. *Identifying query selection strategies that can significantly improve the order of regret remains an open problem in the conversational bandit literature.*

In this work, we address the computational challenges and limited theoretical guarantees of EVOI, as well as the marginal regret improvements in existing conversational bandits, by integrating EVOI into the conversational bandit framework. EVOI and conversational bandits complement and improve each other: EVOI provides an effective query selection strategy, while conversational bandits offer theoretical guarantees on long-term performance. Specifically, we employ two key techniques. First, we propose a novel *gradient-based EVOI* method that replaces the complex Bayesian posterior updates with efficient stochastic gradient descent, directly leveraging the linear reward structure of conversational bandits (details in Section 4.1). This approach significantly reduces computational complexity and facilitates our theoretical analysis. Second, we introduce the concept of *smoothed key term contexts*, inspired by the smoothed analysis framework [30], where queries are randomly perturbed to enhance exploration (details in Section 4.2). For example, instead of querying about "jazz", perturbations like "smooth jazz" or "jazz fusion" can help uncover more specific preferences in users' musical tastes. We show that the combination of these two key techniques is not only intuitive but also has *strong theoretical implications*, leading to substantial and provable improvements in cumulative regret. We incorporate our methods into both Bayesian and frequentist variants of conversational bandits, proposing two new algorithms: ConTS-EVOI, based on conversational Thompson sampling (ConTS [24]), and ConUCB-EVOI, based on conversational LinUCB (ConUCB [41]). Through theoretical analysis, we demonstrate that our methods achieve improved regret bounds compared to existing conversational bandit algorithms (details in Section 6).

To the best of our knowledge, our contributions are threefold: First, we are the first to demonstrate the theoretical implications of integrating EVOI into bandit learning in terms of cumulative regret, establishing a new understanding of how EVOI can enhance query efficiency in conversational recommender systems. Second, we provide a novel and practical query selection strategy that substantially improves regret bounds, advancing the state of the art in conversational bandit literature. Third, our work bridges the gap between two previously separate lines of research, potentially leading to new research opportunities.

In summary, our contributions are listed as follows.

- **Innovative Conversational Mechanism**: We propose a novel conversational mechanism for conversational recommender systems that consists of two key techniques: *gradient-based EVOI* and *smoothed key term contexts*. Gradient-based EVOI provides a computationally efficient criterion for query selection, while smoothed key term contexts enhance exploration by introducing random perturbations to queries.
- **New Algorithms and Theoretical Improvements**: We integrate our conversational mechanism into conversational bandits and propose two new algorithms, ConTS-EVOI and ConUCB-EVOI. We prove that ConTS-EVOI achieves a regret bound of $\widetilde{O}(d\sqrt{T})$ and ConUCB-EVOI achieves $\widetilde{O}(\sqrt{dT} + d)$, both offering a $\sqrt{d}$ improvement in their dependence on $T$ over prior studies.
- **Extensive Empirical Evaluations**: We conduct comprehensive evaluations on both synthetic and real-world datasets. Our algorithms ConTS-EVOI and ConUCB-EVOI consistently outperform baseline methods, reducing regret by over 8.5% and 10.7%, respectively, and validating the effectiveness of our approach.

## 2 Related Work

The problem of query selection and recommendation in conversational recommender systems has been extensively studied in both the recommender systems and online learning communities. The primary focus has been on devising effective query selection strategies to better elicit user preferences. Two orthogonal lines of research have emerged to address this problem.

The first line of research is known as *preference elicitation* [21], which focuses on *how to select queries* to efficiently elicit user preferences. A key concept in this area is the Expected Value of Information (EVOI) [18], which provides a principled criterion for selecting queries and determining when to make recommendations within Bayesian settings. EVOI selects queries that maximize the expected quality (i.e., reward) of the posterior decision (i.e., recommendation) based on possible user responses. Several studies have explored the application of EVOI for query selection. For example, Guo and Sanner [16] select high-EVOI queries by assuming a Gaussian distribution over rewards. Viappiani and Boutilier [35] propose an approximate iterative algorithm for optimizing EVOI, making the computation more tractable. Vendrov et al. [34] introduce a continuous formulation of EVOI as a differentiable network and optimize it using gradient-based methods. Martin et al. [26] develop model-free variants of EVOI that rely on function approximation to obviate specific modeling assumptions. Despite its conceptual appeal, EVOI faces significant computational challenges even for simple distributions. To mitigate the computational issue, some studies employ alternative approaches with similar underlying ideas, such as maximum information gain [7, 44], entropy-based methods [1],

polyhedral/volumetric methods [19], ellipsoidal methods [28], and minimax-regret methods [6]. Our proposed *gradient-based EVOI* aligns with these studies in that it retains the core idea of the original EVOI but enhances computational efficiency by replacing the Bayesian posterior updates with incremental updates using stochastic gradient descent. This adjustment not only addresses the computational challenges but also facilitates theoretical analysis.

The second line of research focuses on *conversational contextual bandits*, initially introduced by Zhang et al. [41]. This framework enables recommenders to launch conversations and obtain feedback on "key terms" to accelerate user preference elicitation. Since then, numerous follow-up studies have extended this framework, exploring various enhancements both empirically and theoretically. For example, CtoF-ConUCB [37] introduces clustering to automatically learn key terms and reduce human labeling efforts. RelativeConUCB [39] proposes comparison-based interactions to get comparative feedback from users. GraphConUCB [43] integrates knowledge graphs as additional information and uses D-optimal design to select key terms. Hier-UCB and Hier-LinUCB [45] leverage the hierarchical structure between key terms and items to improve recommendation efficiency. DecUCB [38] employs causal inference to handle biases resulting from items and key terms. ConLinUCB [36] proposes computing the *barycentric spanner* of the key terms as an efficient exploration basis. FedConPE [25] studies conversational bandits under federated settings and uses an adaptive approach for initiating conversations. However, most of these studies extend the ConUCB algorithm proposed by Zhang et al. [41] without introducing significant theoretical improvements. In contrast, our work presents a novel conversational mechanism that achieves substantially improved regret bounds.

## 3 Background and Problem Formulation

In this section, we introduce the problem settings of conversational bandits and present the original form of the Expected Value of Information (EVOI). In the following, we use boldface letters for vectors and matrices. We denote $[M] := \{1, \ldots, M\}$ for $M \in \mathbb{N}^+$. For any real vector $\boldsymbol{x}, \boldsymbol{y}$ and positive semi-definite (PSD) matrix $\boldsymbol{V}$, $\|\boldsymbol{x}\|$ denotes the $\ell_2$-norm of $\boldsymbol{x}$, and $\|\boldsymbol{x}\|_{\boldsymbol{V}}$ denotes the Mahalanobis norm $\sqrt{\boldsymbol{x}^\mathsf{T} \boldsymbol{V} \boldsymbol{x}}$. We use $\lambda_{\min}(\cdot)$ and $\lambda_{\max}(\cdot)$ to denote the minimum and maximum eigenvalue.

### 3.1 Conversational Bandits

Following prior work [13, 24, 36, 41], we model the conversational multi-armed bandit as a sequential decision-making problem. At each time step $t \in [T]$, the learner (i.e., recommender system) receives a set of arms $\mathcal{A}_t$ (i.e., recommendable items), where each arm $a \in \mathcal{A}_t$ is associated with a *known* feature vector $\boldsymbol{x}_a \in \mathbb{R}^d$. We denote the set of feature vectors as $\{\boldsymbol{x}_a\}_{a \in \mathcal{A}_t} \subseteq \mathbb{R}^d$, which constitutes the *context* at time $t$. Then the learner selects an arm $a_t \in \mathcal{A}_t$ (i.e., recommend an item) and receives a reward $r_t \in \mathbb{R}$ from the environment (i.e., whether the user clicks on the item). We assume the reward follows a linear structure: $r_t = \boldsymbol{x}_{a_t}^\mathsf{T} \boldsymbol{\theta}^* + \eta_t$, where $\boldsymbol{\theta}^* \in \Theta \subseteq \mathbb{R}^d$ is a *fixed* but *unknown* preference vector that the learner aims to learn, and $\eta_t$ is zero-mean, 1-sub-Gaussian noise. The objective is to design a learning policy that minimizes the cumulative regret $R(T)$, defined as the difference between the cumulative rewards of the optimal policy and the learner's policy:

$$R(T) = \sum_{t=1}^{T} \left( \max_{a \in \mathcal{A}_t} \boldsymbol{x}_a^\mathsf{T} \boldsymbol{\theta}^* - \boldsymbol{x}_{a_t}^\mathsf{T} \boldsymbol{\theta}^* \right). \tag{1}$$

In addition to pulling arms (i.e., recommending items), the learner can also occasionally engage in conversations with the user by querying on *key terms* and obtain feedback to accelerate the elicitation of user preferences. Specifically, a *key term* refers to a keyword or topic associated with a subset of arms (e.g., the key term "programming language" may relate to arms like "C/C++", "Python", "Java"). Let $\mathcal{K}$ represent the finite set of key terms, with each element $k \in \mathcal{K}$ associated with a *known* feature vector $\boldsymbol{x}_k \in \mathbb{R}^d$. At time $t$, the learner can choose to query a key term $k_t \in \mathcal{K}$, and receive user feedback modeled as $\widetilde{r}_t = \boldsymbol{x}_{k_t}^\mathsf{T} \boldsymbol{\theta}^* + \widetilde{\eta}_t$, where $\boldsymbol{\theta}^*$ is the same user preference vector as in the arm selection and $\widetilde{\eta}_{i,t}$ is zero-mean, 1-sub-Gaussian noise. It is important to note that in the literature, the preference vectors for arms and key terms are often assumed to be either very close or identical. In our setting, we follow the formulation of Li et al. [24, 25], Wang et al. [36] and assume that they are identical. Following the standard assumptions in bandit literature [22], without loss of generality, we assume that feature vectors for both arms and key terms are normalized, i.e., $\|\boldsymbol{x}_a\| = \|\boldsymbol{x}_k\| = 1$ for all $a \in \mathcal{A}_t$ and $k \in \mathcal{K}$. We also assume that the unknown preference vector is bounded, i.e., $\|\boldsymbol{\theta}^*\| \leq 1$.

### 3.2 Bayesian Preference Elicitation

In the Bayesian approach of recommendation and preference elicitation, the recommender system maintains a *probability distribution* over the unknown user preference vector $\boldsymbol{\theta}^*$ and updates this distribution as more information is gathered through interactions with the user. While a fully Bayesian approach uses this distribution to generate both recommendations and elicitation queries, alternative criteria may be preferred for computational efficiency consideration. Here we introduce the general concepts and notations following previous studies [34]. Detailed constructions and computations of our gradient-based method will be presented in Section 4.

The system starts with a prior distribution $P_0$ over the unknown preference vector $\boldsymbol{\theta}^*$ defined on $\Theta$, which reflects the system's initial belief about the user's preferences. It is worth noting that the choice of $P_0$ does not affect our theoretical results and can be derived based on past interactions with other users in practice. As the recommender system interacts with the user and collects feedback, the prior distribution is updated to form the current belief, denoted by $P$. Mathematically, both $P_0$ and $P$ are probability density functions over the preference vector space $\Theta$. Given the current belief $P$, the *expected reward* of an arm $a \in \mathcal{A}_t$ is defined as:

$$\mathrm{ER}(a; P) = \mathbb{E}_{\boldsymbol{\theta} \sim P}[\boldsymbol{x}_a^\mathsf{T} \boldsymbol{\theta}] = \boldsymbol{x}_a^\mathsf{T} \mathbb{E}_{\boldsymbol{\theta} \sim P}[\boldsymbol{\theta}] = \boldsymbol{x}_a^\mathsf{T} \int_{\boldsymbol{\theta} \in \Theta} \boldsymbol{\theta} P(\boldsymbol{\theta}) \, \mathrm{d}\boldsymbol{\theta}.$$

At each round $t$, the optimal arm given the current belief $P$ is the arm that maximizes the expected reward:

$$a_{t,P}^* = \arg\max_{a \in \mathcal{A}_t} \mathrm{ER}(a; P); \quad \mathrm{ER}_t^*(P) = \mathrm{ER}(a_{t,P}^*; P).$$

For each key term $k \in \mathcal{K}$, let $\mathcal{R}_k$ be the set of possible user responses (e.g., "yes" or "no"). The recommender system has a response model that specifies the probability $Q(r \mid k, \boldsymbol{\theta})$ that a user with preference

$\boldsymbol{\theta}$ responds $r \in \mathcal{R}_k$ when queried with key term $k \in \mathcal{K}$. Under the current belief $P$, the expected probability of observing response $r$ when querying key term $k$ is $Q(r \mid k; P) = \mathbb{E}_{\boldsymbol{\theta} \sim P}[Q(r \mid k, \boldsymbol{\theta})]$. After receiving response $r$ to key term $k$, the posterior belief $Q(\boldsymbol{\theta} \mid r, k)$ (denoted as $Q_r^k(\boldsymbol{\theta})$ for convenience) is updated using Bayes' rule:

$$Q_r^k(\boldsymbol{\theta}) \triangleq Q(\boldsymbol{\theta} \mid r, k) = \frac{Q(r \mid k, \boldsymbol{\theta}) P(\boldsymbol{\theta})}{\int_{\boldsymbol{\theta} \in \Theta} Q(r \mid k, \boldsymbol{\theta}) P(\boldsymbol{\theta}) \, d\boldsymbol{\theta}}. \tag{2}$$

### 3.3 Expected Value of Information

We focus on a myopic (greedy) query selection strategy known as the *Expected Value of Information (EVOI)* [9, 18], which employs a one-step look-ahead to select queries that maximize the expected reward. Specifically, after querying a key term $k$ at time $t$ given the current belief $P$, the *posterior expected reward* can be written as:

$$\begin{aligned}
\text{PER}_t(k; P) &= \sum_{r \in \mathcal{R}_k} Q(r \mid k; P) \text{ER}^*(Q_r^k) \\
&= \sum_{r \in \mathcal{R}_k} Q(r \mid k; P) \max_{a \in \mathcal{A}_t} \boldsymbol{x}_a^\top \int_{\boldsymbol{\theta} \in \Theta} \boldsymbol{\theta} Q_r^k(\boldsymbol{\theta}) \, d\boldsymbol{\theta} \\
&= \max_{a \in \mathcal{A}_t} \boldsymbol{x}_a^\top \underbrace{\int_{\boldsymbol{\theta} \in \Theta} \boldsymbol{\theta} \sum_{r \in \mathcal{R}_k} Q(r \mid k; P) Q_r^k(\boldsymbol{\theta}) \, d\boldsymbol{\theta}}_{\triangleq \boldsymbol{\theta}_P^k}, \tag{3}
\end{aligned}$$

where $\boldsymbol{\theta}_P^k$ represents the expected preference vector after querying key term $k$ based on belief $P$, accounting for all possible responses weighted by their probabilities. Intuitively, $\text{PER}_t(k; P)$ is the expected reward of the best possible recommendation *after* querying key term $k$ and updating the belief based on the user's potential responses. The *expected value of information (EVOI)* is then defined as the difference between the posterior expected reward and the expected reward without additional information:

$$\text{EVOI}_t(k; P) = \text{PER}_t(k; P) - \text{ER}_t^*(P). \tag{4}$$

The EVOI measures the expected *improvement* in the maximum reward achievable by the recommender system after querying key term $k$, relative to the current belief $P$. Therefore, by selecting a key term that maximizes $\text{EVOI}_t(k; P)$ (which also maximizes the posterior expected reward $\text{PER}_t(k; P)$), the learner gains the most valuable information that will lead to better recommendations.

## 4 Key Techniques

In this section, we introduce the key techniques used in our algorithms. First, we present a novel *gradient-based EVOI* method, inspired by the incremental update in stochastic gradient descent. This method leverages the linear reward structure to avoid the computationally expensive Bayesian posterior updates via Bayes' rule. Second, we introduce the concept of *smoothed key term contexts*, following insights from Kannan et al. [20], Raghavan et al. [27], where random perturbations are added to key term contexts to accelerate the exploration of user preferences.

### 4.1 Gradient-based EVOI

While the conventional EVOI introduced in Section 3 is theoretically sound, its Bayesian updating process is computationally intensive, especially in high-dimensional settings, and can be unnecessary

given the linear reward structure in conversational bandits. To address this, inspired by the incremental updates of stochastic gradient descent (SGD), we propose *gradient-based EVOI*, which eliminates the need for Bayesian updates as in Equation (2). Specifically, instead of maintaining probability distributions $P$ and $Q$ to model the preference vector and query responses as outlined in Section 3.2, we maintain two vectors $\boldsymbol{\theta}_t^{\text{prior}}$ and $\widehat{\boldsymbol{\theta}}_t$. Here, $\boldsymbol{\theta}_t^{\text{prior}}$ represents the prior estimate of the user's preference vector at the beginning of round $t$, and $\widehat{\boldsymbol{\theta}}_t$ represents the updated belief after incorporating any user feedback received during round $t$.

Recall that in stochastic gradient descent for linear regression, the estimated parameter is updated as:

$$\boldsymbol{\theta}_{t+1} = \boldsymbol{\theta}_t - \alpha \nabla L(\boldsymbol{\theta}_t) = \boldsymbol{\theta}_t - \alpha(\boldsymbol{x}^\top \boldsymbol{\theta}_t - y)\boldsymbol{x},$$

where $\alpha$ is the learning rate, $\nabla L(\boldsymbol{\theta}_t)$ is the gradient of the loss function at $\boldsymbol{\theta}_t$, and $(\boldsymbol{x}, y)$ is the newly observed data point. Inspired by this, and similar to the calculation of $\boldsymbol{\theta}_P^k$ in Equation (3), we simulate the potential update to our preference estimate that would result from querying key term $k$ based on the current estimate. Since we have not yet observed the user's actual response, the same as the calculation of $\boldsymbol{\theta}_P^k$, we consider the expected feedback based on our current estimate: $r = \boldsymbol{x}_k^\top \widehat{\boldsymbol{\theta}}_t$. We then simulate the updated preference vector after querying key term $k$ based on the prior estimate $\boldsymbol{\theta}_t$ by computing:

$$\widetilde{\boldsymbol{\theta}}_t^k = \boldsymbol{\theta}_t - \alpha(\boldsymbol{x}_k^\top \boldsymbol{\theta}_t - r)\boldsymbol{x}_k. \tag{5}$$

Then similar to Equation (3), the posterior expected reward at time $t$ is computed as $\text{PER}_t(k) = \max_{a \in \mathcal{A}_t} \boldsymbol{x}_a^\top \widetilde{\boldsymbol{\theta}}_t^k$. And we select the key term that maximizes it:

$$k_t = \arg\max_{k \in \mathcal{K}} \max_{a \in \mathcal{A}_t} \boldsymbol{x}_a^\top \widetilde{\boldsymbol{\theta}}_t^k. \tag{6}$$

In Table 1, we summarize the notations used in the original EVOI and our gradient-based EVOI, highlighting their one-to-one correspondence for better understanding. In Section 5, we apply gradient-based EVOI to our algorithms, and in Section 6, we show that the simplicity of this method facilitates our theoretical analysis and leads to tighter regret bounds.

**Table 1: Comparison between EVOI and gradient-based EVOI.**

| Meaning | EVOI | Gradient-based EVOI |
|---|---|---|
| Current belief/estimate of $\boldsymbol{\theta}^*$ | $P$ | $\boldsymbol{\theta}_t^{\text{prior}}$ |
| Belief/estimate after query feedback | $Q$ | $\widehat{\boldsymbol{\theta}}_t$ |
| Expected preference vector after query | $\boldsymbol{\theta}_P^k$ | $\widetilde{\boldsymbol{\theta}}_t^k$ |
| Posterior expected reward | $\max_{a \in \mathcal{A}_t} \boldsymbol{x}_a^\top \boldsymbol{\theta}_P^k$ | $\max_{a \in \mathcal{A}_t} \boldsymbol{x}_a^\top \widetilde{\boldsymbol{\theta}}_t^k$ |

### 4.2 Smoothed Key Term Contexts

To better explore users' preferences and facilitate our theoretical analysis, we introduce the concept of *smoothed key term contexts*. This idea is inspired by the smoothed adversary setting, originally introduced by Kannan et al. [20], which studies how greedy algorithms behave in multi-armed bandits. In our context, smoothed key term contexts allow for a more nuanced exploration of user preferences by adding random perturbations to the feature vectors associated with key terms. The goal is to extend the exploration

space, thus revealing a richer set of user preferences. For example, instead of using the key term "jazz", perturbations like "smooth jazz", "jazz fusion", or "bebop" could help uncover more specific preferences in users' musical tastes. This randomness facilitates exploration beyond what a static set of key terms might capture, ultimately improving user modeling and personalization. The perturbations are formally modeled as Gaussian noise as follows.

**Definition 1** (Smoothed key term contexts). Given a set of key terms $\mathcal{K}$ and their associated feature vectors $\{x_k\}_{k \in \mathcal{K}}$, define the *smoothed feature vector* for each key term $k \in \mathcal{K}$ as $\widetilde{x}_k = x_k + \varepsilon_k$, where $\varepsilon_k \in \mathbb{R}^d$ is a noise vector independently sampled from a truncated multivariate Gaussian distribution with mean zero and covariance matrix $\sigma^2 I$, and each dimension of the noise is truncated within the interval $[-R, R]$. That is, $\varepsilon_k \sim \mathcal{N}(0, \sigma^2 I)$ subject to $|(\varepsilon_k)_j| \leq R, \forall j \in [d]$. We refer to the resulting set of perturbed feature vectors $\{\widetilde{x}_k\}_{k \in \mathcal{K}}$ as the *smoothed key term context*.

## 5 Algorithm Design

In this section, we apply gradient-based EVOI and smoothed key term contexts to both conversational linear Thompson sampling (Bayesian approach) and conversational LinUCB (frequentist approach) based on ConTS [24] and ConUCB [41].

The general workflow of our algorithms is illustrated in Figure 1. When the recommender system initiates conversations with the user, it generates smoothed key term contexts and selects appropriate key terms based on gradient-based EVOI. The feedback received during the conversation, along with item feedback, is then integrated into the recommendation process. For decision-making on items, the system (learner) utilizes either a Bayesian approach such as Thompson Sampling (TS) or a frequentist approach like Upper Confidence Bound (UCB) to recommend items to the user.

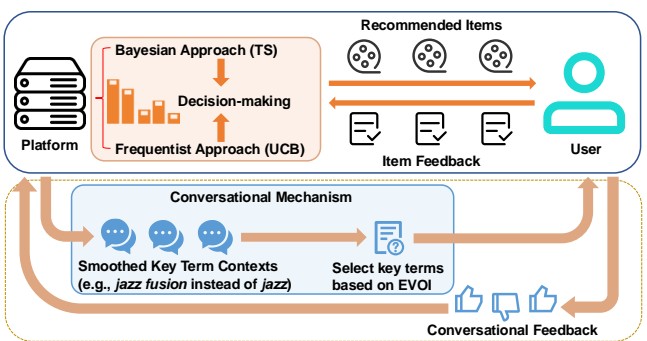

**Figure 1: The general workflow of our algorithm design.**

### 5.1 Conversational LinTS with EVOI

As shown in Algorithm 1, at each round $t = 1, 2, \ldots$, the learner receives a set of recommendable items (arms) $\mathcal{A}_t$ and the algorithm operates in two decision-making phases: key term selection (i.e., selecting queries, Lines 5 to 11) and arm selection (i.e., recommending items, Lines 12 to 16). To determine when and how many queries to initiate, we follow Zhang et al. [41] and define an increasing function $b : \mathbb{N}^+ \mapsto \mathbb{R}^+$ to control the conversation frequency. At

each round $t$, if $q_t = \lfloor b(t) \rfloor - \lfloor b(t-1) \rfloor > 0$, the recommender is permitted to conduct $q_t$ conversational queries (Line 5); otherwise, no conversation is initiated.

---

**Algorithm 1:** ConTS-EVOI

**Input:** $\lambda, \delta, \alpha, \beta_t, \sigma, R, b(t)$
**Init:** $V_1 = \lambda I_{d \times d}, b_1 = 0_{d \times 1}, \widehat{\theta}_1 = 0_{d \times 1}, \Sigma_1 = \beta_1^2 V_1^{-1}$

1 **for** $t = 1, 2, \ldots$ **do**
2    Receive arm set $\mathcal{A}_t$
3    $\theta_t^{\text{prior}} = \widehat{\theta}_t$
4    $q_t = \lfloor b(t) \rfloor - \lfloor b(t-1) \rfloor$
5    **while** $q_t > 0$ **do**
6      Generate smoothed key term context $\{\widetilde{x}_k\}_{k \in \mathcal{K}}$
7      Let $\widetilde{\theta}_t^k \triangleq \theta_t^{\text{prior}} - \alpha(\widetilde{x}_k^\top \theta_t^{\text{prior}} - \widetilde{x}_k^\top \widehat{\theta}_t)\widetilde{x}_k$ for all $k \in \mathcal{K}$
8      Select key term $k_t = \arg\max_{k \in \mathcal{K}} \max_{a \in \mathcal{A}_t} x_a^\top \widetilde{\theta}_t^k$
9      Receive key term-level reward $\widetilde{r}_t$
10      Update statistics:
$$V_t = V_t + \widetilde{x}_{k_t}\widetilde{x}_{k_t}^\top, \quad b_t = b_t + \widetilde{r}_t\widetilde{x}_{k_t}, \quad \widehat{\theta}_t = V_t^{-1}b_t$$
11      $q_t = q_t - 1$
12    Draw $\theta_t^{\text{TS}} \sim \mathcal{N}(\widehat{\theta}_t, \Sigma_t)$
13    Select arm $a_t = \arg\max_{a \in \mathcal{A}_t} x_a^\top \theta_t^{\text{TS}}$
14    Receive reward $r_t$
15    Update statistics:
$$V_{t+1} = V_t + x_{a_t}x_{a_t}^\top, \quad b_{t+1} = b_t + r_t x_{a_t}, \quad \widehat{\theta}_{t+1} = V_{t+1}^{-1}b_{t+1}$$
16    Compute covariance matrix: $\Sigma_{t+1} = \beta_{t+1}^2 V_{t+1}^{-1}$

---

*5.1.1 Key Term Selection.* When the algorithm is allowed to initiate conversations, it first stores the current preference estimate $\widehat{\theta}_t$ as the prior estimate $\theta_t^{\text{prior}}$ (Line 3). Then it generates the smoothed key term context by adding random perturbations to each key term $k \in \mathcal{K}$ according to Definition 1 (Line 6). Note that the smoothed context is regenerated every time a conversation is initiated. Here we slightly abuse the notation and always use $\{\widetilde{x}_k\}_{k \in \mathcal{K}}$ to denote the smoothed key term contexts across different conversations. The algorithm computes the gradient-based EVOI for each key term using Equation (5) (Line 7) and selects the key term $k_t$ that maximizes the posterior expected reward as in Equation (6) (Line 8). After receiving user feedback $\widetilde{r}_t$ on the key term $k_t$, the algorithm updates the preference estimate $\widehat{\theta}_t$ using ridge regression (Line 10).

*5.1.2 Arm Selection.* In the arm selection phase, ConTS-EVOI employs the Thompson sampling algorithm [32] to select arms for recommendation. Following existing studies [3, 4], we assume that the user's true preference vector $\theta^*$ follows a multivariate Gaussian distribution $\mathcal{N}(\widehat{\theta}_t, \Sigma_t)$, where $\widehat{\theta}_t$ denotes the estimated mean of $\theta^*$ and $\Sigma_t$ denotes the covariance matrix capturing the uncertainty in the estimate. The algorithm proceeds by sampling $\theta_t^{\text{TS}}$ from the posterior distribution (Line 12) to get an estimate of the preference vector with uncertainty. Then it selects the arm $a_t$ that maximizes the expected reward under the sampled preference vector $\theta_t^{\text{TS}}$ (Line 13). Finally, upon receiving the reward $r_t$ from recommending $a_t$, the

algorithm updates the preference estimate $\widehat{\theta}_t$ using ridge regression (Line 15) and recalculates the covariance matrix $V_t$ to reflect the updated uncertainty (Line 16). The parameter $\beta_t$ in Line 16 is defined according to our theoretical analysis as follows:

$$\beta_t = \sqrt{2\log(\frac{1}{\delta}) + d\log\left(1 + \left(t + (1 + \sqrt{d}R)b(t)\right)/\lambda d\right)} + \sqrt{\lambda},$$

where $\delta \in (0, 1)$ is a confidence parameter, and $\lambda$ is the regularization parameter used in ridge regression to calculate $\widehat{\theta}_t$.

## 5.2 Conversational LinUCB with EVOI

As shown in Algorithm 2, the `ConUCB-EVOI` algorithm follows a similar structure to `ConTS-EVOI`. The key difference lies in the arm selection phase, where `ConUCB-EVOI` employs the LinUCB algorithm [2] to select arms for recommendation. Specifically, it maintains a confidence ellipsoid around the estimated preference vector $\widehat{\theta}_t$ and selects the arm $a_t$ with the highest *upper confidence bound* at each round $t$ (Line 12). Then based on the observed reward $r_t$, the algorithm updates the covariance matrix $V_t$ and the preference estimate $\widehat{\theta}_t$ using ridge regression (Line 14).

---

**Algorithm 2:** ConUCB-EVOI

**Input:** $\lambda$, $\delta$, $\alpha$, $\beta_t$, $\sigma$, $R$, $b(t)$
**Init:** $V_1 = \lambda I_{d \times d}$, $b_1 = 0_{d \times 1}$, $\widehat{\theta}_t = 0_{d \times 1}$

1   **for** $t = 1, 2, \ldots$ **do**
2     Receive arm set $\mathcal{A}_t$
3     $\theta_t^{\text{prior}} = \widehat{\theta}_t$
4     $q_t = \lfloor b(t) \rfloor - \lfloor b(t-1) \rfloor$
5     **while** $q_t > 0$ **do**
6       Generate smoothed key term context $\{\widetilde{x}_k\}_{k \in \mathcal{K}}$
7       Let $\widetilde{\theta}_t^k \triangleq \theta_t^{\text{prior}} - \alpha(\widetilde{x}_k^\mathsf{T}\theta_t^{\text{prior}} - \widetilde{x}_k^\mathsf{T}\widehat{\theta}_t)\widetilde{x}_k$ for all $k \in \mathcal{K}$
8       Select key term $k_t = \arg\max_{k \in \mathcal{K}} \max_{a \in \mathcal{A}_t} x_a^\mathsf{T}\widetilde{\theta}_t^k$
9       Receive key term-level reward $\widetilde{r}_t$
10      Update statistics:
         $V_t = V_t + \widetilde{x}_{k_t}\widetilde{x}_{k_t}^\mathsf{T}, \quad b_t = b_t + \widetilde{r}_t\widetilde{x}_{k_t}, \quad \widehat{\theta}_t = V_t^{-1}b_t$
11      $q_t = q_t - 1$
12    Select arm $a_t = \arg\max_{a \in \mathcal{A}_t} x_a^\mathsf{T}\widehat{\theta}_t + \beta_t\|x_a\|_{V_t^{-1}}$
13    Receive reward $r_t$
14    Update statistics:
     $V_{t+1} = V_t + x_{a_t}x_{a_t}^\mathsf{T}, \;\; b_{t+1} = b_t + r_t x_{a_t}, \;\; \widehat{\theta}_{t+1} = V_{t+1}^{-1}b_{t+1}$

---

## 6 Theoretical Analysis

This section presents the theoretical results of the cumulative regrets of our algorithms. The proofs of Theorems 1 and 2 are given in Appendices B and C, respectively.

**Theorem 1** (Regret of ConTS-EVOI). *With probability at least $1 - \delta$, the cumulative regret of* `ConTS-EVOI` *scales in* $O\left(d\sqrt{T}\log(T)\right)$.

**Theorem 2** (Regret of ConUCB-EVOI). *With probability at least $1-\delta$, the cumulative regret of* `ConUCB-EVOI` *scales in* $O\left(\sqrt{dT\log(T)} + d\right)$.

---

**Table 2: Comparison of theoretical regret bounds.**

| Strategy | Algorithms | Conversational | Regret |
|---|---|:---:|---|
| Thompson Sampling (TS) | LinTS [3, 4] | ✗ | $\widetilde{O}(d^{\frac{3}{2}}\sqrt{T})$ |
| | ConTS [24] | ✓ | $\widetilde{O}(d^{\frac{3}{2}}\sqrt{T})^*$ |
| | ConTS-EVOI (Ours, Theorem 1) | ✓ | $\widetilde{O}(d\sqrt{T})$ |
| UCB | LinUCB [2] | ✗ | $\widetilde{O}(d\sqrt{T})$ |
| | ConUCB [41], ConLinUCB [36] | ✓ | $\widetilde{O}(d\sqrt{T})$ |
| | ConUCB-EVOI (Ours, Theorem 2) | ✓ | $\widetilde{O}(\sqrt{dT} + d)$ |

$^*$ The original ConTS paper does not present the regret but it can be easily derived.

We summarize the regret bounds for our proposed algorithms and compare them with related algorithms in Table 2, using big-$\widetilde{O}$ notation to suppress logarithmic factors for clearer comparison. We categorize the algorithms into two groups based on their arm selection strategies: Thompson Sampling-based algorithms and LinUCB-based algorithms. As shown in the table, existing algorithms for conversational bandits (`ConTS`, `ConUCB`, `ConLinUCB`) do not achieve substantial improvements over their non-conversational counterparts (`LinTS`, `LinUCB`). This limitation is primarily because they do not fully leverage the additional information obtained from conversations. In contrast, our algorithms employ a novel query selection strategy that enhances exploration and effectively integrates user feedback from queries into the learning process. Consequently, the regret bounds of `ConTS-EVOI` and `ConUCB-EVOI` both have a $\sqrt{d}$ improvement in their dependence on the time horizon $T$, where $T$ may be very large in practice. This improvement highlights the effectiveness of our approach in utilizing conversational queries to enhance recommendation quality.

## 7 Evaluation

In this section, we conduct extensive experiments to demonstrate the effectiveness of our proposed algorithms. Specifically, we aim to answer the following research questions:

(1) Do our algorithms `ConTS-EVOI` and `ConUCB-EVOI` outperform existing state-of-the-art conversational bandit algorithms?
(2) How does the additional information obtained from conversational queries affect the estimation of user preferences and overall performance?
(3) What is the impact of the conversation frequency function $b(t)$ on performance?

### 7.1 Experiment Settings

*7.1.1 Datasets.* Following previous studies [12, 38, 39, 41, 43], we generate a synthetic dataset and use the following real-world datasets.

- **MovieLens-25M [17]**: The MovieLens-25M dataset is collected from the movie recommendation website MovieLens[1]. It contains 25,000,095 ratings, created by 162,541 users for 62,423 movies.
- **Last.fm [8]**: The Last.fm dataset is collected from the online music platform Last.fm[2]. It contains 186,479 tag assignments that link 1,892 users with 17,632 artists.
- **Yelp[3]**: The Yelp dataset is collected by Yelp, a platform where users contribute reviews and ratings for various businesses such

---

[1] https://movielens.org/    [2] https://www.last.fm    [3] https://www.yelp.com/dataset

as restaurants and shops. It contains 6,990,280 reviews for 150,346 businesses created by 1,987,897 users.

*7.1.2  Data Generation and Preprocessing.* Our synthetic dataset is generated following the methodology used in previous studies [25, 36, 37, 41]. We set the feature dimension $d = 50$, the number of users (i.e., the number of unknown preference vectors) $N = 200$, the number of arms $|\mathcal{A}| = 5000$, and the number of key terms $|\mathcal{K}| = 1000$. To reflect the relationship between key terms and arms, we generate the dataset as follows. We first sample $|\mathcal{K}|$ pseudo feature vectors $\{\dot{x}_k\}_{k \in \mathcal{K}}$, where each component of $\dot{x}_k$ is independently drawn from the uniform distribution $\mathcal{U}(-1, 1)$. Then for each arm $a$, we uniformly sample a subset $\mathcal{K}_a \subset \mathcal{K}$ consisting of $n_a$ key terms, where $n_a$ is a random integer selected uniformly from $\{1, 2, \ldots, 5\}$. We assign a weight $w_{a,k} = 1/n_a$ for each key term $k \in \mathcal{K}_a$. Finally, the feature vector for each arm $a$, denoted by $x_a$, is drawn from a multivariate Gaussian $\mathcal{N}(\sum_{j \in \mathcal{K}_a} \dot{x}_j / n_a, I)$. And the feature vector for each key term $k$, denoted by $x_k$, is computed as $x_k = \sum_{a \in \mathcal{A}} \frac{w_{a,k}}{\sum_{j \in \mathcal{A}} w_{j,k}} x_a$. This models the idea that each key term is related to a few arms. Each user $u \in [N]$ is represented as a preference vector $\theta_u \in \mathbb{R}^d$. We generate $N$ preference vectors by sampling each component of $\theta_u$ independently from $\mathcal{U}(-1, 1)$.

For the real-world datasets, we consider movies/artists/businesses as arms. To facilitate data analysis, we extract a subset of $|\mathcal{A}| = 5,000$ arms with the highest number of user-assigned ratings/tags, and a subset of $N = 200$ users who have provided the most ratings/tags. Key terms are identified using the associated metadata: movie genres, business categories, or tag IDs in the MovieLens, Yelp, and Last.fm datasets, respectively. For example, each movie is associated with a list of genres, such as "Action", "Comedy", or "Drama", and each business (e.g., restaurant) has a list of categories, such as "Burgers", "Seafood", or "Steakhouses". Using the data extracted above, we create a *feedback matrix* $R$ of size $N \times |\mathcal{A}|$, where each element $R_{i,j}$ represents the user $i$'s feedback to arm $j$. We assume that the user's feedback is binary. For the MovieLens and Yelp datasets, $R_{i,j} = 1$ if user $i$'s rating for arm $j$ is higher than 3; otherwise, $R_{i,j} = 0$. For the Last.fm dataset, $R_{i,j} = 1$ if user $i$ assigns a tag to artist $j$; otherwise, $R_{i,j} = 0$.

Next, we generate the feature vectors for arms $\{x_a\}_{a \in \mathcal{A}}$ and the preference vectors for users $\{\theta_u\}_{u \in [N]}$ following the existing works [36, 37, 43]. We decompose the feedback matrix $R$ using Singular Value Decomposition (SVD) as $R = \Theta S A^\mathsf{T}$, where $\Theta = \{\theta_u\}_{u \in [N]}$, and $A = \{x_a\}_{a \in \mathcal{A}}$. Then we extract the top $d = 50$ dimensions of these vectors associated with the highest singular values in $S$. The feature vectors for key terms $\{x_k\}_{k \in \mathcal{K}}$ are generated following Zhang et al. [41], which maintains equal weights for all key terms corresponding to each arm.

*7.1.3  Baseline Algorithms.* We select the following algorithms from existing studies as baselines for comparison with our methods.

- LinUCB [2, 23]: The standard linear contextual bandit algorithm designed for *infinite* arm sets. It does not consider conversational interactions and relies only on arm-level feedback.
- LinTS [4]: The standard Thompson sampling algorithm for linear contextual bandits, also utilizing only arm-level feedback.
- ConUCB [41]: The original algorithm proposed for conversational contextual bandits. It initiates conversations by querying key

terms when allowed and leverages the feedback from these queries to accelerate the learning of user preferences.
- ConLinUCB [36]: A series of algorithms that modify the key term selection strategy of ConUCB. It contains three variants: ConLinUCB-BS computes the *barycentric spanner* of key terms to form an efficient exploration basis. ConLinUCB-MCR chooses key terms with the maximum confidence radius. ConLinUCB-UCB chooses key terms with the largest upper confidence bound.

*7.1.4  Performance Metrics.* To measure the performance of our algorithms, we employ two metrics. First, we use the cumulative regret defined in Equation 1, which is a standard metric in bandit literature. A lower cumulative regret signifies better performance. Second, we evaluate the estimation error of the unknown preference vector by computing the $\ell_2$-norm of the difference between the estimated vector $\widehat{\theta}_t$ and the true preference vector $\theta^*$. This metric quantifies how closely the algorithm's estimate aligns with the user's actual preferences at each time step. To ensure statistical reliability, we repeat each experiment 20 times and report the average results. The confidence intervals derived from these repetitions are also included in the figures. All the experiments were conducted on a MacBook Pro with M1 Pro 8-core CPU and 16GB of RAM.

## 7.2  Evaluation Results

*7.2.1  Cumulative Regret.* we evaluate and compare our proposed algorithms, ConTS-EVOI and ConUCB-EVOI, against all the baseline algorithms in terms of cumulative regret. The evaluation is conducted over $T = 4,000$ rounds. In each round, we randomly sample $K = 50$ arms from $\mathcal{A}$ to form the arm set for that round. For generating the smoothed key term contexts, we set both the variance of perturbation $\sigma^2$ and the truncation limit $R$ to 1. For computing gradient-based EVOI, the learning rate $\alpha$ is set to 0.1. Following existing studies [36, 41], we adopt the conversation frequency function $b(t) = 5\lfloor \log(t) \rfloor$. The results are presented in Figure 2, where the $x$-axis is the number of rounds and the $y$-axis is the cumulative regret. Among all the datasets, we observe consistent trends that align with the findings from previous studies. Specifically, all the algorithms exhibit sublinear growth in cumulative regret over time, indicating effective learning and adaptation. Algorithms that do not utilize key term queries (i.e., LinUCB and LinTS) show the poorest performance (highest cumulative regret). In contrast, algorithms that incorporate conversational interactions by querying key terms perform significantly better, showing the importance of conversations in accelerating preference elicitation. Our algorithms ConTS-EVOI and ConUCB-EVOI consistently achieve the best performance, reducing cumulative regret by more than 8.5% and 10.7%, respectively, compared to the best baseline. This demonstrates the effectiveness of our novel conversational mechanism by integrating gradient-based EVOI and smoothed key term contexts into the conversational bandit framework.

*7.2.2  Accuracy of Estimated Preference Vectors.* In Figure 3, we present the average distance between the estimated vector $\widehat{\theta}_t$ and the ground truth $\theta^*$ for all algorithms from rounds $t = 100$ to 900, illustrating how accurately each algorithm learns the user's preferences over time. It is important to note that although all algorithms initiate conversations simultaneously and with the same

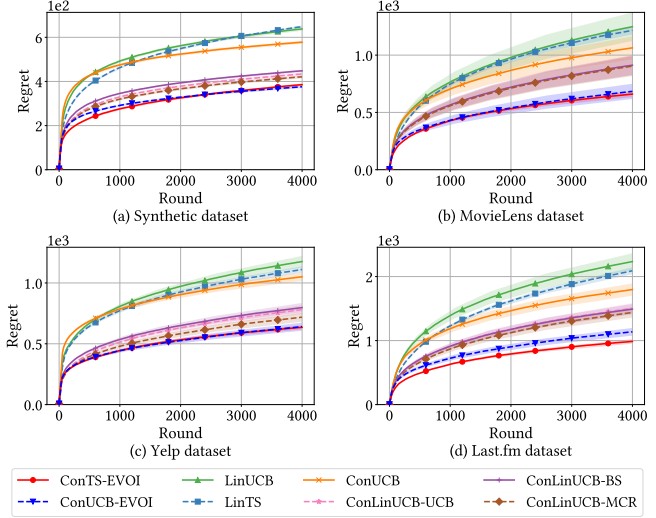

Figure 2: Comparison of cumulative regret.

frequency (since they use the same conversation frequency function $b(t)$), only our algorithms ConTS-EVOI and ConUCB-EVOI exhibit distinct *staircase-like curves* in the figure. Each steep drop in these curves corresponds to a round of conversations. This pattern demonstrates that, by leveraging our novel conversational mechanism, ConTS-EVOI and ConUCB-EVOI can gather more informative feedback during the conversation, thus sharply decreasing the estimation error, whereas other algorithms do not show such advantages. As a result, our algorithms estimate the user's preference vector more quickly and accurately than the baseline algorithms.

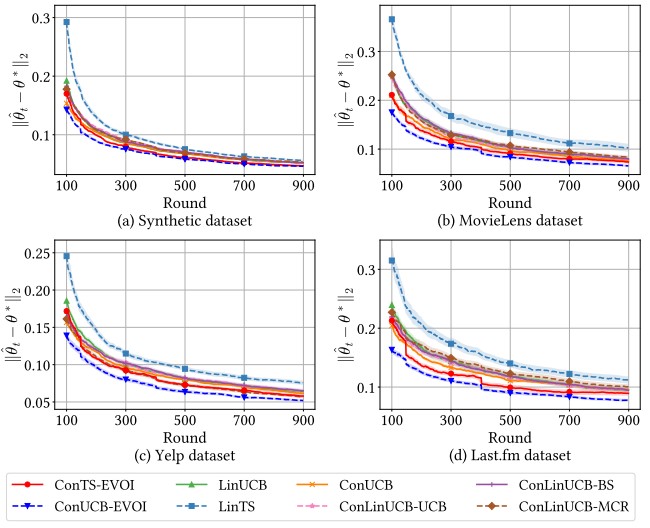

Figure 3: Comparison of estimation error.

*7.2.3 Influence of Conversation Frequency.* We evaluate the impact of the conversation frequency function $b(t)$, which determines how often conversations occur as the number of rounds $t$ increases.

Similar to Zhang et al. [41], we employ a series of conversation frequency functions defined as $b(t) = n \cdot \lfloor \log(t) \rfloor$, where $n$ is the number of questions asked during each round of conversation, and the interval between consecutive conversations grows exponentially. We measure the cumulative regret and the cumulative estimation error $\|\widehat{\theta}_t - \theta^*\|_2$ over $T = 1,000$ rounds. Due to space constraints, we present results only for the real-world Last.fm dataset; other datasets exhibit similar patterns. As illustrated in Figures 4 and 5, for all the conversational algorithms, the cumulative regret and estimation error consistently decrease with increasing $n$, underscoring the benefits of more frequent conversations. Notably, ConTS-EVOI and ConUCB-EVOI show the most significant improvements as $n$ increases, indicating that they benefit more from frequent conversations. This suggests that our algorithms are more effective at leveraging conversational information than the baseline algorithms.

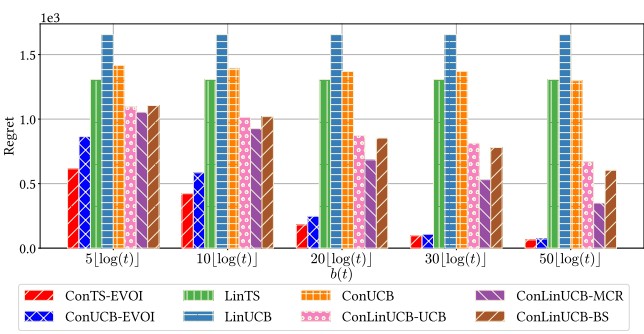

Figure 4: Cumulative regret with different $b(t)$.

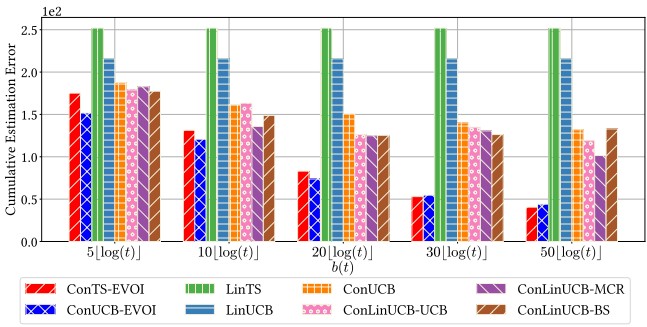

Figure 5: Cumulative estimation error with different $b(t)$.

## 8 Conclusion

In this paper, we bridged the gap between two previously separate lines of research by incorporating the expected value of information (EVOI) into the conversational bandit framework. We introduced a novel conversational mechanism built on two key techniques: *gradient-based EVOI* and *smoothed key term contexts*. Based on these techniques, we proposed two new algorithms, ConTS-EVOI and ConUCB-EVOI, and proved that they achieve significantly tighter regret bounds than existing approaches. Our extensive evaluations further confirmed that our algorithms outperform current state-of-the-art conversational bandit algorithms.

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

## A Technical Lemmas for Gradient-based EVOI and Smoothed Key Term Contexts

**Lemma 1.** *For any round $t \in [T]$, with the smoothed key term contexts in Definition 1, gradient-based EVOI key term selection strategy has the following lower bound on the minimum eigenvalue of the matrix $\mathbb{E}[\widetilde{x}_{k_t}\widetilde{x}_{k_t}^{\mathsf{T}}]$ for the selected key term $k_t \in \mathcal{K}$:*

$$\lambda_{\min}\left(\mathbb{E}\left[\widetilde{x}_k\widetilde{x}_k^{\mathsf{T}}\right]\right) \geq c_1 \frac{\sigma^2}{\log|\mathcal{K}|} \triangleq \lambda_{\mathcal{K}},$$

*where $c_1 \in (0,1)$ is some constant.*

PROOF. Fix a time $t$, and denote the key term selected at this time as $k_t$. Note that although multiple key terms may be selected at each time step, they all satisfy this lemma. Therefore, we do not distinguish between them and use only a single subscript $t$. For each arm $a$, let $Q_a$ be a unitary matrix that rotates the feature vector $x_a$ to align it with the $x$-axis, maintaining its length but zeroing out all components except the first component, i.e., $Q_a x_a = (\|x_a\|, 0, 0, \ldots, 0)$. Such matrices $\{Q_a\}_{a \in \mathcal{A}_t}$ always exist because they just rotate the space. Recall the key term selection of gradient-based EVOI in Equation (6): $k_t = \arg\max_{k \in \mathcal{K}} \max_{a \in \mathcal{A}_t} x_a^{\mathsf{T}}\widetilde{\theta}_t^k$, where $\widetilde{\theta}_t^k = \theta_t^{\text{prior}} - \alpha(\widetilde{x}_k^{\mathsf{T}}\theta_t^{\text{prior}} - \widetilde{x}_k^{\mathsf{T}}\widehat{\theta}_t)\widetilde{x}_k$. For simplicity, we also define a shorthand $\alpha_t^k \triangleq -\alpha(\widetilde{x}_k^{\mathsf{T}}\theta_t^{\text{prior}} - \widetilde{x}_k^{\mathsf{T}}\widehat{\theta}_t)$ and $y_t^k \triangleq \theta_t^{\text{prior}} + \alpha_t^k x_k$. We have

$$\lambda_{\min}\left(\mathbb{E}\left[\widetilde{x}_{k_t}\widetilde{x}_{k_t}^{\mathsf{T}}\right]\right) = \lambda_{\min}\left(\mathbb{E}\left[xx^{\mathsf{T}} \,\Big|\, x = \arg\max_{\widetilde{x}_k:k\in\mathcal{K}} \max_{a\in\mathcal{A}_t} x_a^{\mathsf{T}}\widetilde{\theta}_t^k\right]\right)$$

$$= \min_{w:\|w\|=1} w^{\mathsf{T}} \mathbb{E}\left[xx^{\mathsf{T}} \,\Big|\, x = \arg\max_{\widetilde{x}_k:k\in\mathcal{K}} \max_{a\in\mathcal{A}_t} x_a^{\mathsf{T}}\widetilde{\theta}_t^k\right] w$$

$$= \min_{w:\|w\|=1} \mathbb{E}\left[(w^{\mathsf{T}}x)^2 \,\Big|\, x = \arg\max_{\widetilde{x}_k:k\in\mathcal{K}} \max_{a\in\mathcal{A}_t} x_a^{\mathsf{T}}\widetilde{\theta}_t^k\right]$$

$$\geq \min_{w:\|w\|=1} \text{Var}\left[w^{\mathsf{T}}x \,\Big|\, x = \arg\max_{\widetilde{x}_k:k\in\mathcal{K}} \max_{a\in\mathcal{A}_t} x_a^{\mathsf{T}}\widetilde{\theta}_t^k\right]$$

$$= \min_{w:\|w\|=1} \text{Var}\left[w^{\mathsf{T}}x \,\Big|\, x = \arg\max_{\widetilde{x}_k:k\in\mathcal{K}} \max_{a\in\mathcal{A}_t} x_a^{\mathsf{T}}(\theta_t^{\text{prior}} + \alpha_t^k\widetilde{x}_k)\right]$$

$$= \min_{w:\|w\|=1} \text{Var}\left[w^{\mathsf{T}}x \,\Big|\, x = \arg\max_{\widetilde{x}_k:k\in\mathcal{K}} \max_{a\in\mathcal{A}_t} x_a^{\mathsf{T}}(y_t^k + \alpha_t^k\varepsilon_k)\right] \quad (7)$$

$$= \min_{w:\|w\|=1} \text{Var}\left[w^{\mathsf{T}}x \,\Big|\, x = \arg\max_{\widetilde{x}_k:k\in\mathcal{K}} \max_{a\in\mathcal{A}_t} (Q_a x_a)^{\mathsf{T}} Q_a(y_t^k + \alpha_t^k\varepsilon_k)\right] \quad (8)$$

$$= \min_{w:\|w\|=1} \text{Var}\left[w^{\mathsf{T}}x \,\Big|\, x = \arg\max_{\widetilde{x}_k:k\in\mathcal{K}} \max_{a\in\mathcal{A}_t} \|x_a\|(Q_a(y_t^k + \alpha_t^k\varepsilon_k))_1\right] \quad (9)$$

$$= \min_{w:\|w\|=1} \text{Var}\left[w^{\mathsf{T}}\varepsilon \,\Big|\, \varepsilon = \arg\max_{\varepsilon_k:k\in\mathcal{K}} \left[(Q_a\varepsilon_k)_1 + \max_{a\in\mathcal{A}_t}\left(\frac{Q_a}{\alpha_t^k}y_t^k\right)_1\right]\right] \quad (10)$$

$$= \min_{w:\|w\|=1} \text{Var}\left[w^{\mathsf{T}}\varepsilon \,\Big|\, \varepsilon = \arg\max_{\varepsilon_k:k\in\mathcal{K}} \left[(\varepsilon_k)_1 + \max_{a\in\mathcal{A}_t}\left(\frac{Q_a}{\alpha_t^k}y_t^k\right)_1\right]\right] \quad (11)$$

where Equation (7) uses the definition of smoothed key term $\widetilde{x}_k = x_k + \varepsilon_k$. Equation (8) uses the property of unitary matrices: $Q_a^{\mathsf{T}}Q_a = I_d$. Equation (9) applies matrix $Q_a$ so only the first component is non-zero. Equation (10) follows because the randomness of a

smoothed key term $\widetilde{x}_k = x_k + \varepsilon_k$ stems from $\varepsilon_k$, and adding a constant $a$ to a random variable does not change its variance. Equation (11) is due to the rotation invariance of symmetrically truncated Gaussian distributions.

Since $\varepsilon_k \sim \mathcal{N}(0, \sigma^2 I_d)$ conditioned on $|(\varepsilon_k)_j| \leq R, \forall j \in [d]$, by the property of (truncated) multivariate Gaussian distributions, the components of $\varepsilon_k$ can be equivalently regarded as $d$ independent samples from a (truncated) univariate Gaussian, i.e., $(\varepsilon_k)_j \sim \mathcal{N}(0, \sigma^2)$ conditioned on $|(\varepsilon_k)_j| \leq R, \forall j \in [d]$. Therefore, we have

$$\text{Var}\left[w^{\mathsf{T}}\varepsilon\right] = \text{Var}\left[\sum_{i=1}^d w_i(\varepsilon)_i\right] = \sum_{i=1}^d w_i^2 \text{Var}\left[(\varepsilon)_i\right],$$

where the exchanging of variance and summation is due to the independence of $(\varepsilon)_i$. Therefore, by denoting $z_t^k = \max_{a\in\mathcal{A}_t}\left(\frac{Q_a}{\alpha_t^k}y_t^k\right)_1$, we can write

$$\min_{w:\|w\|=1} \text{Var}\left[w^{\mathsf{T}}\varepsilon \,\Big|\, \varepsilon = \arg\max_{\varepsilon_k:k\in\mathcal{K}}\left((\varepsilon_k)_1 + z_t^k\right)\right]$$

$$= \min_{w:\|w\|=1} \sum_{j=1}^d w_j^2 \text{Var}\left[(\varepsilon)_j \,\Big|\, \varepsilon = \arg\max_{\varepsilon_k:k\in\mathcal{K}}\left((\varepsilon_k)_1 + z_t^k\right)\right]$$

$$= \min_{w:\|w\|=1} \left\{ w_1^2 \text{Var}\left[(\varepsilon)_1 \,\Big|\, \varepsilon = \arg\max_{\varepsilon_k:k\in\mathcal{K}}\left((\varepsilon_k)_1 + z_t^k\right)\right] \right.$$
$$\left. + \sum_{j=2}^d w_j^2 \text{Var}\left[(\varepsilon)_j \,\Big|\, \varepsilon = \arg\max_{\varepsilon_k:k\in\mathcal{K}}\left((\varepsilon_k)_1 + z_t^k\right)\right]\right\}$$

$$= \min_{w:\|w\|=1} \left\{ w_1^2 \text{Var}\left[(\varepsilon)_1 \,\Big|\, \varepsilon = \arg\max_{\varepsilon_k:k\in\mathcal{K}}\left((\varepsilon_k)_1 + z_t^k\right)\right] + \sum_{j=2}^d w_j^2 \text{Var}\left[(\varepsilon)_j\right]\right\}$$

$$= \min_{w:\|w\|=1} \left\{ w_1^2 \text{Var}\left[(\varepsilon)_1 \,\Big|\, \varepsilon = \arg\max_{\varepsilon_k:k\in\mathcal{K}}\left((\varepsilon_k)_1 + z_t^k\right)\right] + (1 - w_1^2)\sigma^2\right\}$$

$$= \min\left\{ \text{Var}\left[(\varepsilon)_1 \,\Big|\, \varepsilon = \arg\max_{\varepsilon_k:k\in\mathcal{K}}\left((\varepsilon_k)_1 + z_t^k\right)\right], \sigma^2\right\} \geq c_1 \frac{\sigma^2}{\log|\mathcal{K}|},$$

where in the last inequality, we use Lemma 15 and Lemma 14 in Sivakumar et al. [29] and get

$$\text{Var}\left[(\varepsilon)_1 \,\Big|\, \varepsilon = \arg\max_{\varepsilon_k:k\in\mathcal{K}}\left((\varepsilon_k)_1 + z_t^k\right)\right]$$
$$\geq \text{Var}\left[(\varepsilon)_1 \,\Big|\, \varepsilon = \arg\max_{\varepsilon_k:k\in\mathcal{K}}(\varepsilon_k)_1\right] \geq c_1 \frac{\sigma^2}{\log|\mathcal{K}|}.$$

□

**Lemma 2.** *Let $b(t) = bt$ for some $b \in (0,1)$. With probability at least $1 - \delta$ for any $\delta \in (0,1)$, if $t \geq T_0 \triangleq \frac{8(1+\sqrt{d}R)^2}{b\lambda_{\mathcal{K}}}\log\left(\frac{d}{\delta}\right)$, we have*

$$\lambda_{\min}\left(\sum_{s=1}^t \sum_{k\in\mathcal{K}_s} \widetilde{x}_k\widetilde{x}_k^{\mathsf{T}}\right) \geq \frac{\lambda_{\mathcal{K}}bt}{2}.$$

*where $\lambda_{\mathcal{K}} = c_1 \frac{\sigma^2}{\log|\mathcal{K}|}$ is defined in Lemma 1, and $\mathcal{K}_t$ denotes the set of key terms queried at round $t$.*

PROOF. To apply the matrix Chernoff bound (Lemma 6), we first verify the required two conditions for the self-adjoint matrices $\widetilde{x}_k \widetilde{x}_k^\top$ for any $k \in \mathcal{K}_s$ and $s \in [t]$. First, $\widetilde{x}_k \widetilde{x}_k^\top$ is obviously positive semi-definite. Second, by the Courant-Fischer theorem,

$$\lambda_{\max}(\widetilde{x}_k \widetilde{x}_k^\top) = \max_{w:\|w\|=1} w^\top \widetilde{x}_k \widetilde{x}_k^\top w = \max_{w:\|w\|=1} (w^\top \widetilde{x}_k)^2$$

$$\leq \max_{w:\|w\|=1} \|w\|^2 \|\widetilde{x}_k\|^2 \leq (1+\sqrt{d}R)^2.$$

Next, by Lemma 1 and the super-additivity of the minimum eigenvalue (due to Weyl's inequality), we have

$$\mu_{\min} = \lambda_{\min}\left(\sum_{s=1}^{t} \sum_{k \in \mathcal{K}_s} \mathbb{E}\left[\widetilde{x}_k \widetilde{x}_k^\top\right]\right) \geq \sum_{s=1}^{t} \sum_{k \in \mathcal{K}_s} \lambda_{\min}\left(\mathbb{E}\left[\widetilde{x}_k \widetilde{x}_k^\top\right]\right) \geq \lambda_{\mathcal{K}} bt,$$

where the last inequality is because there are at most $bt$ key terms selected by round $t$, so the summation has at most $bt$ terms. So by Lemma 6, we have for any $\varepsilon \in (0,1)$,

$$\Pr\left[\lambda_{\min}\left(\sum_{s=1}^{t} \sum_{k \in \mathcal{K}_s} \widetilde{x}_k \widetilde{x}_k^\top\right) \leq (1-\varepsilon)\lambda_{\mathcal{K}} bt\right]$$

$$\leq \Pr\left[\lambda_{\min}\left(\sum_{s=1}^{t} \sum_{k \in \mathcal{K}_s} \widetilde{x}_k \widetilde{x}_k^\top\right) \leq (1-\varepsilon)\mu_{\min}\right]$$

$$\leq d\left[\frac{e^{-\varepsilon}}{(1-\varepsilon)^{1-\varepsilon}}\right]^{\mu_{\min}/(1+\sqrt{d}R)^2} \leq d\left[\frac{e^{-\varepsilon}}{(1-\varepsilon)^{1-\varepsilon}}\right]^{\frac{\lambda_{\mathcal{K}} bt}{(1+\sqrt{d}R)^2}},$$

where the last inequality is because $e^{-x}$ is decreasing. Choosing $\varepsilon = \frac{1}{2}$, we get

$$\Pr\left[\lambda_{\min}\left(\sum_{s=1}^{t} \sum_{k \in \mathcal{K}_s} \widetilde{x}_k \widetilde{x}_k^\top\right) \leq \frac{\lambda_{\mathcal{K}} bt}{2}\right] \leq d\left(\sqrt{2}e^{-\frac{1}{2}}\right)^{\frac{\lambda_{\mathcal{K}} bt}{(1+\sqrt{d}R)^2}}.$$

Letting the RHS be $\delta$, we get $t = \frac{2(1+\sqrt{d}R)^2 \log(\frac{d}{\delta})}{\lambda_{\mathcal{K}} b(1-\log(2))} \leq \frac{8(1+\sqrt{d}R)^2}{\lambda_{\mathcal{K}} b} \log\left(\frac{d}{\delta}\right)$. Therefore, $\lambda_{\min}\left(\sum_{s=1}^{t} \sum_{k \in \mathcal{K}_s} \widetilde{x}_k \widetilde{x}_k^\top\right) \geq \frac{\lambda_{\mathcal{K}} bt}{2}$ holds with probability at least $1 - \delta$ when $t \geq \frac{8(1+\sqrt{d}R)^2}{\lambda_{\mathcal{K}} b} \log\left(\frac{d}{\delta}\right)$. $\square$

## B Regret Analysis of Algorithm 1

First, we introduce some notations to facilitate our analysis.

Recall that $\beta_t = \sqrt{2 \log\left(\frac{1}{\delta}\right) + d \log\left(1 + \frac{t+\left(1+\sqrt{d}R\right)b(t)}{\lambda d}\right)} + \sqrt{\lambda}$. For each time step $t$, we define the event $\mathcal{E}_t^\theta$ as follows:

$$\mathcal{E}_t^\theta = \left\{\left|x_a^\top \widehat{\theta}_t - x_a^\top \theta^*\right| \leq \beta_t \|x_a\|_{V_t^{-1}}, \forall a \in \mathcal{A}_t\right\}.$$

Let $a_t^* = \arg\max_{a \in \mathcal{A}_t} x_a^\top \theta^*$ be the optimal arm at time step $t$. The mean reward gap between the optimal arm and any arm $a \in \mathcal{A}_t$ is given by $\Delta_t^a = x_{a_t^*}^\top \theta^* - x_a^\top \theta^*$.

We define a filtration $\{\mathcal{F}_t\}_{t \geq 0}$, where $\mathcal{F}_{t-1}$ captures all relevant information up to time step $t - 1$ and includes the contexts of all arms at time step $t$, i.e.,

$$\mathcal{H}_{t-1} = \left\{\{x_a\}_{a \in \mathcal{A}_s}, a_s, r_s, \widetilde{\mathcal{D}}_s, \mathcal{K}_s, \{\widetilde{r}_{s,k}\}_{k \in \mathcal{K}_s}, s \leq t-1, \{x_a\}_{a \in \mathcal{A}_t}\right\},$$

where $\mathcal{K}_s$ denotes the set of key terms queried at time step $s$ and $\{\widetilde{r}_{s,k}\}_{k \in \mathcal{K}_s}$ denotes the corresponding rewards from pulling the key terms in $\mathcal{K}_s$. $\mathcal{K}_s = \emptyset$ if no key term is queried at time step $s$.

Next, we present some lemmas used in the proof of Theorem 1.

**Lemma 3** (Lemma 4 in Agrawal and Goyal [4]). *For any $t \geq 1$, for any $\mathcal{F}_{t-1}$ such that the event $\mathcal{E}_t^\theta$ holds, we have*

$$\mathbb{E}[\Delta_t^{a_t} \mid \mathcal{F}_{t-1}] \leq G_t \widetilde{\beta}_t \mathbb{E}[\|x_{a_t}\|_{V_t^{-1}} \mid \mathcal{F}_{t-1}] + \frac{2}{t^2},$$

*where $G_t$ and $\widetilde{\beta}_t$ are defined as $G_t = \left(\frac{2}{\frac{1}{4e\sqrt{\pi}} - \frac{1}{t^2}} + 1\right)$ and $\widetilde{\beta}_t = (2\sqrt{d \log t} + 1)\beta_t$ respectively.*

**Lemma 4.** *For any $\delta \in (0,1)$, for all $t \geq 1$ and all arm $a \in \mathcal{A}_t$, with probability at least $1 - \delta$, we have*

$$\left|x_a^\top \widehat{\theta}_t - x_a^\top \theta^*\right| \leq \beta_t \|x_a\|_{V_t^{-1}}.$$

Lemma 4 is equivalent to that the event $\mathcal{E}_t^\theta$ holds for all $t \geq 1$ with probability at least $1 - \delta$. It demonstrates that the estimated reward of each arm is close to the true reward with high probability.

PROOF. Before selecting arm $a_t$ at time $t$, $\widehat{\theta}_t = V_t^{-1} b_t$, and

$$V_t = \sum_{s=1}^{t-1} x_{a_s} x_{a_s}^\top + \sum_{s=1}^{t} \sum_{k \in \mathcal{K}_s} \widetilde{x}_k \widetilde{x}_k^\top + \lambda I_{d \times d},$$

$$b_t = \sum_{s=1}^{t-1} r_s x_{a_s} + \sum_{s=1}^{t} \sum_{k \in \mathcal{K}_s} \widetilde{r}_{s,k} \widetilde{x}_k.$$

For any $a \in \mathcal{A}_t$, we have

$$x_a^\top \widehat{\theta}_t - x_a^\top \theta^* = x_a^\top \left(V_t^{-1} b_t - \theta^*\right)$$

$$= x_a^\top \left(V_t^{-1}\left(\sum_{s=1}^{t-1} r_s x_{a_s} + \sum_{s=1}^{t} \sum_{k \in \mathcal{K}_s} \widetilde{r}_{s,k} \widetilde{x}_k\right) - \theta^*\right)$$

$$= x_a^\top \left(V_t^{-1}\left(\sum_{s=1}^{t-1} x_{a_s}\left(x_{a_s}^\top \theta^* + \eta_s\right) + \sum_{s=1}^{t} \sum_{k \in \mathcal{K}_s} \widetilde{x}_k\left(\widetilde{x}_k^\top \theta^* + \widetilde{\eta}_s\right)\right) - \theta^*\right)$$

$$= \lambda x_a^\top V_t^{-1} \theta^* + x_a^\top \left(V_t^{-1}\left(\sum_{s=1}^{t-1} x_{a_s} \eta_s + \sum_{s=1}^{t} \sum_{k \in \mathcal{K}_s} \widetilde{x}_k \widetilde{\eta}_s\right)\right). \quad (12)$$

For the first term, we have

$$|\lambda x_a^\top V_t^{-1} \theta^*| = \lambda \|x_a\|_{V_t^{-1}} \|\theta^*\|_{V_t^{-1}}$$

$$\leq \lambda \|x_a\|_{V_t^{-1}} \frac{\|\theta^*\|}{\sqrt{\lambda}} = \sqrt{\lambda} \|x_a\|_{V_t^{-1}}, \quad (13)$$

where the inequality follows that $\|\theta^*\|_{V_t^{-1}}^2 \leq \frac{\|\theta^*\|^2}{\lambda_{\min}(V_t)}$ and $\lambda_{\min}(V_t) \geq \lambda$, and the last equality follows the assumption that $\|\theta^*\| = 1$.

According to Theorem 1 in Abbasi-Yadkori et al. [2], for any $\delta \in (0,1)$, with probability at least $1 - \delta$, for all $t \geq 1$, we have

$$\left\|\sum_{s=1}^{t-1} x_{a_s} \eta_s + \sum_{s=1}^{t} \sum_{k \in \mathcal{K}_s} \widetilde{x}_k \widetilde{\eta}_s\right\|_{V_t^{-1}} \leq \sqrt{2 \log\left(\frac{\det(V_t)^{1/2} \det(\lambda I_{d \times d})^{-1/2}}{\delta}\right)}.$$

Besides, we have

$$\text{tr}(V_t) \leq d\lambda + \sum_{s=1}^{t-1} \text{tr}(x_{a_s} x_{a_s}^\top) + \sum_{s=1}^{t} \sum_{k \in \mathcal{K}_s} \text{tr}(\widetilde{x}_k \widetilde{x}_k^\top)$$

$$\leq d\lambda + t + \left(1 + \sqrt{d}R\right) b(t),$$

where $\text{tr}(A)$ denotes the trace of matrix $A$, The last inequality follows from the fact that $\|\widetilde{x}_k\| \leq 1 + \sqrt{d}R$ for all $k \in \mathcal{K}_s$ by Definition 1, and there are at most $b(t)$ key terms selected by $t$.

Then based on the determinant-trace inequality (Lemma 7),

$$\det(V_t) \leq \left(\frac{\text{tr}(V_t)}{d}\right)^d \leq \left(\frac{d\lambda + t + \left(1 + \sqrt{d}R\right)b(t)}{d}\right)^d.$$

Therefore, we have

$$\left\| \sum_{s=1}^{t-1} x_{a_s} \eta_s + \sum_{s=1}^{t} \sum_{k \in \mathcal{K}_s} \widetilde{x}_k \widetilde{\eta}_s \right\|_{V_t^{-1}}$$

$$\leq \sqrt{2\log\left(\frac{1}{\delta}\right) + d\log\left(1 + \frac{t + \left(1 + \sqrt{d}R\right)b(t)}{\lambda d}\right)}. \quad (14)$$

Applying Cauchy-Schwarz inequality to Equation (12) and plugging Equations (13) and (14) into it, we have

$$|x_a^\top \widehat{\theta}_t - x_a^\top \theta^*|$$

$$\leq \lambda |x_a^\top V_t^{-1} \theta^*| + \|x_a\|_{V_t^{-1}} \left\| \sum_{s=1}^{t-1} x_{a_s} \eta_s + \sum_{s=1}^{t} \sum_{k \in \mathcal{K}_s} \widetilde{x}_k \widetilde{\eta}_s \right\|_{V_t^{-1}}$$

$$\leq \|x_a\|_{V_t^{-1}} \left( \sqrt{\lambda} + \sqrt{2\log\left(\frac{1}{\delta}\right) + d\log\left(1 + \frac{t + \left(1 + \sqrt{d}R\right)b(t)}{\lambda d}\right)} \right)$$

$$= \beta_t \|x_a\|_{V_t^{-1}},$$

where the last inequality follows the definition of $\beta_t$. $\square$

Following Wang et al. [36], we choose $b(t) = bt$ for some $b \in (0, 1)$. We now present the following lemma and prove Theorem 1.

**Lemma 5.** When $t \geq T_0 \triangleq \frac{8(1+\sqrt{d}R)^2}{b\lambda_{\mathcal{K}}} \log\left(\frac{d}{\delta}\right)$, for any $a \in \mathcal{A}_t$, with probability at least $1 - \delta$ for some $\delta \in (0, 1)$, we have $\|x_a\|_{V_t^{-1}} \leq \sqrt{\frac{2}{\lambda_{\mathcal{K}} bt}}$.

PROOF. Note that for any $a \in \mathcal{A}_t$,

$$\|x_a\|_{V_t^{-1}} = \sqrt{x_a^\top V_t^{-1} x_a} \leq \sqrt{\lambda_{\max}(V_t^{-1}) x_a^\top x_a} = \sqrt{\frac{1}{\lambda_{\min}(V_t)}}, \quad (15)$$

where the inequality is due to the matrix operator norm, and the last equality follows from the assumption that $\|x_a\| = 1$.

According to the definition of $V_t$ in Algorithm 1, we have

$$\lambda_{\min}(V_t) = \lambda_{\min}\left(\sum_{s=1}^{t-1} x_{a_s} x_{a_s}^\top + \sum_{s=1}^{t} \sum_{k \in \mathcal{K}_s} \widetilde{x}_k \widetilde{x}_k^\top + \lambda I_{d \times d}\right)$$

$$\geq \lambda_{\min}\left(\sum_{s=1}^{t} \sum_{k \in \mathcal{K}_s} \widetilde{x}_k \widetilde{x}_k^\top\right) \geq \frac{\lambda_{\mathcal{K}} bt}{2}, \quad (16)$$

where the last inequality follows from Lemma 2.

Combining Equations (15) and (16), we know $\|x_a\|_{V_t^{-1}} \leq \sqrt{\frac{2}{\lambda_{\mathcal{K}} bt}}$. $\square$

**Theorem 1** (Regret of ConTS-EVOI). *With probability at least $1 - \delta$, the cumulative regret of* ConTS-EVOI *scales in* $O\left(d\sqrt{T} \log(T)\right)$.

PROOF OF THEOREM 1. We define the instantaneous regret at time $t$ as $\text{reg}_t = \Delta_t^{a_t} = x_{a_t^*}^\top \theta^* - x_{a_t}^\top \theta^*$, and denote $\text{reg}_t' = \text{reg}_t \mathbb{1}[\mathcal{E}_t^\theta]$, where $\mathbb{1}[\mathcal{E}_t^\theta]$ is the indicator function that takes the value 1 if the event $\mathcal{E}_t^\theta$ holds and 0 otherwise.

Define $Y_0 = 0$ and $Y_t = \sum_{s=1}^{t} \text{reg}_s' - G_s \widetilde{\beta}_s \|x_{a_s}\|_{V_s^{-1}} - \frac{2}{s^2}$ for all $t \geq 1$. We first show that $\{Y_t\}_{t \geq 0}$ is a super-martingale with respect to the filtration $\{\mathcal{F}_t\}_{t \geq 0}$, i.e., $\mathbb{E}[Y_t - Y_{t-1} \mid \mathcal{F}_{t-1}] \leq 0$ for all $t \geq 1$.

When the event $\mathcal{E}_t^\theta$ holds, we have $\text{reg}_t' = \text{reg}_t$, then

$$\mathbb{E}[Y_t - Y_{t-1} \mid \mathcal{F}_{t-1}] = \mathbb{E}[\text{reg}_t' - G_t \widetilde{\beta}_t \|x_{a_t}\|_{V_t^{-1}} - \frac{2}{t^2} \mid \mathcal{F}_{t-1}]$$

$$\leq G_t \widetilde{\beta}_t \mathbb{E}[\|x_{a_t}\|_{V_t^{-1}} \mid \mathcal{F}_{t-1}] + \frac{2}{t^2} - G_t \widetilde{\beta}_t \mathbb{E}[\|x_{a_t}\|_{V_t^{-1}} \mid \mathcal{F}_{t-1}] - \frac{2}{t^2}$$

$$\leq 0,$$

where the first inequality follows from Lemma 3.

When the event $\mathcal{E}_t^\theta$ does not hold, we have $\text{reg}_t' = 0$, then the inequality $\mathbb{E}[Y_t - Y_{t-1} \mid \mathcal{F}_{t-1}] \leq 0$ holds naturally.

Besides, $|Y_t - Y_{t-1}|$ is bounded as follows:

$$|Y_t - Y_{t-1}| \leq \text{reg}_t' + G_t \widetilde{\beta}_t \|x_{a_t}\|_{V_t^{-1}} + \frac{2}{t^2}$$

$$\leq 4 + G\widetilde{\beta}_T \frac{1}{\sqrt{\lambda}} \triangleq B,$$

where the second inequality follows from the fact that $\text{reg}_t' \leq 2$, $\|x_{a_t}\|_{V_t^{-1}} \leq \lambda_{\min}^{-1/2}(V_t)\|x_{a_t}\| \leq \frac{1}{\sqrt{\lambda}}$, $G \triangleq \max_{t=1,2,\dots} |G_t|$ is a constant, and $\widetilde{\beta}_t$ is increasing in $t$. We define $B$ as the upper bound of $|Y_t - Y_{t-1}|$ for all $t \geq 1$.

Recall that $T_0 = \frac{8(1+\sqrt{d}R)^2}{b\lambda_{\mathcal{K}}} \log\left(\frac{d}{\delta}\right)$. Applying Azuma-Hoeffding inequality (Lemma 8) to the super-martingale $\{Y_t\}_{t \geq T_0}$, we know with probability at least $1 - \delta$ it holds that

$$Y_T - Y_{T_0} \leq \sqrt{2\log\left(\frac{1}{\delta}\right) \sum_{t=T_0+1}^{T} |Y_t - Y_{t-1}|^2} \leq B\sqrt{2\log\left(\frac{1}{\delta}\right)T},$$

which implies that with probability at least $1 - \delta$, we have

$$\sum_{t=T_0+1}^{T} \text{reg}_t' \leq B\sqrt{2\log\left(\frac{1}{\delta}\right)T} + \sum_{t=T_0+1}^{T} G_t \widetilde{\beta}_t \|x_{a_t}\|_{V_t^{-1}} + \sum_{t=T_0+1}^{T} \frac{2}{t^2}. \quad (17)$$

By Lemma 5, we have that with probability at least $1 - \delta$,

$$\sum_{t=T_0+1}^{T} G_t \widetilde{\beta}_t \|x_{a_t}\|_{V_t^{-1}} \leq G\widetilde{\beta}_T \sum_{t=T_0+1}^{T} \|x_{a_t}\|_{V_t^{-1}}$$

$$\leq G\widetilde{\beta}_T \sqrt{\frac{2}{\lambda_{\mathcal{K}} b}} \sum_{t=T_0+1}^{T} \frac{1}{\sqrt{t}}$$

$$\leq 2G\widetilde{\beta}_T \sqrt{\frac{2}{\lambda_{\mathcal{K}} b} T}, \tag{18}$$

where the first inequality follows from that $G \triangleq \max_{t=1,2,\dots} |G_t|$ is a constant and $\widetilde{\beta}_t$ is increasing in $t$, the second inequality follows from Lemma 5, and the last inequality follows from the fact that $\sum_{t=1}^{T} \frac{1}{\sqrt{t}} \leq 2\sqrt{T}$.

Combining Equations (18) and (17), we have that with probability at least $1 - 2\delta$, it holds that

$$\sum_{t=1}^{T} \text{reg}_t' \leq \sum_{t=1}^{T_0} \text{reg}_t' + \sum_{t=T_0+1}^{T} \text{reg}_t'$$

$$\leq \sum_{t=1}^{T_0} \text{reg}_t' + B\sqrt{2\log\left(\frac{1}{\delta}\right)T} + 2G\widetilde{\beta}_T \sqrt{\frac{2}{\lambda_{\mathcal{K}} b} T} + \sum_{t=T_0+1}^{T} \frac{2}{t^2}$$

$$\leq 2T_0 + B\sqrt{2\log\left(\frac{1}{\delta}\right)T} + 2G\widetilde{\beta}_T \sqrt{\frac{2}{\lambda_{\mathcal{K}} b} T} + 4,$$

where the last inequality follows from the fact that $\text{reg}_t' \leq \text{reg}_t \leq 2$ for all $t$ and $\sum_{t=1}^{T} \frac{1}{t^2} \leq 2$.

When $\mathcal{E}_t^{\theta}$ holds for all $t$, we have $\sum_{t=1}^{T} \text{reg}_t = \sum_{t=1}^{T} \text{reg}_t'$. According to Lemma 4, we have that with probability at least $1 - \delta$, it holds that $\sum_{t=1}^{T} \text{reg}_t = \sum_{t=1}^{T} \text{reg}_t'$.

Therefore, with probability at least $1 - 3\delta$, we have

$$R(T) = \sum_{t=1}^{T} \text{reg}_t \leq 2T_0 + B\sqrt{2\log\left(\frac{1}{\delta}\right)T} + 2G\widetilde{\beta}_T \sqrt{\frac{2}{\lambda_{\mathcal{K}} b} T} + 4.$$

Note that $\widetilde{\beta}_T = (2\sqrt{d\log T} + 1)\beta_T = O(d\log T)$, $B = 4 + G\widetilde{\beta}_T \frac{1}{\sqrt{\lambda}}$ is in the order of $O(d\log T)$, and $T_0 = \frac{8(1+\sqrt{d}R)^2}{b\lambda_{\mathcal{K}}} \log\left(\frac{d}{\delta}\right)$ defined in Lemma 5 is independent of $T$. Then, with probability at least $1 - 3\delta$,

$$R(T) = \sum_{t=1}^{T} \text{reg}_t = O(d\sqrt{T}\log T). \qquad \square$$

## C    Regret Analysis of Algorithm 2

We present the proof of Theorem 2 in this section. For convenience, we follow the same notations as in Appendix B.

**Theorem 2** (Regret of ConUCB-EVOI). *With probability at least $1-\delta$, the cumulative regret of* ConUCB-EVOI *scales in* $O\left(\sqrt{dT\log(T)} + d\right)$.

PROOF OF THEOREM 2. We first decompose the instantaneous regret at time $t$ under the event $\mathcal{E}_t^{\theta}$ as follows:

$$\text{reg}_t = x_{a_t^*}^{\top} \theta^* - x_{a_t}^{\top} \theta^*$$

$$= x_{a_t^*}^{\top} \left(\theta^* - \widehat{\theta}_t\right) + \left(x_{a_t^*}^{\top} \widehat{\theta}_t + \beta_t \|x_{a_t^*}\|_{V_t^{-1}}\right) - \beta_t \|x_{a_t^*}\|_{V_t^{-1}}$$

$$- \left(x_{a_t}^{\top} \widehat{\theta}_t + \beta_t \|x_{a_t}\|_{V_t^{-1}}\right) + x_{a_t}^{\top} \left(\widehat{\theta}_t - \theta^*\right) + \beta_t \|x_{a_t}\|_{V_t^{-1}}$$

$$\leq x_{a_t^*}^{\top} \left(\theta^* - \widehat{\theta}_t\right) + x_{a_t}^{\top} \left(\widehat{\theta}_t - \theta^*\right) + \beta_t \|x_{a_t}\|_{V_t^{-1}} \tag{19}$$

$$\leq \beta_t \|x_{a_t^*}\|_{V_t^{-1}} + \beta_t \|x_{a_t}\|_{V_t^{-1}} - \beta_t \|x_{a_t^*}\|_{V_t^{-1}} + \beta_t \|x_{a_t}\|_{V_t^{-1}} \tag{20}$$

$$\leq 2\beta_t \|x_{a_t}\|_{V_t^{-1}},$$

where Equation (19) follows from the definition of UCB, Equation (20) follows from the definition of $\mathcal{E}_t^{\theta}$.

Therefore, according to Lemma 4, with probability at least $1-\delta$, it holds that $\text{reg}_t \leq 2\beta_t \|x_{a_t}\|_{V_t^{-1}}$ for all $t \geq 1$. Then with probability at least $1 - \delta$, it holds that

$$R(T) = \sum_{t=1}^{T_0} \text{reg}_t + \sum_{t=T_0+1}^{T} \text{reg}_t$$

$$\leq 2T_0 + \sum_{t=T_0+1}^{T} 2\beta_t \|x_{a_t}\|_{V_t^{-1}}$$

$$\leq 2T_0 + 2\beta_T \sum_{t=T_0+1}^{T} \sqrt{\frac{2}{\lambda_{\mathcal{K}} b t}} \tag{21}$$

$$\leq 2T_0 + 4\beta_T \sqrt{\frac{2}{\lambda_{\mathcal{K}} b} T}, \tag{22}$$

where Equation (21) follows from Lemma 5, and Equation (22) follows from the fact that $\sum_{t=1}^{T} \frac{1}{\sqrt{t}} \leq 2\sqrt{T}$.

Recall that $T_0 = \frac{8(1+\sqrt{d}R)^2}{b\lambda_{\mathcal{K}}} \log\left(\frac{d}{\delta}\right)$ defined in Lemma 5 and $\beta_T = O(\sqrt{d\log T})$, we have with probability at least $1 - \delta$, it holds that

$$R(T) = O(\sqrt{dT\log T} + d). \qquad \square$$

## D    Technical Inequalities

**Lemma 6** (Matrix Chernoff, Corollary 5.2 in Tropp [33]). *Consider a finite sequence $\{X_k\}$ of independent, random, self-adjoint matrices with dimension $d$. Assume that each random matrix satisfies*

$$X_k \succeq 0 \quad \text{and} \quad \lambda_{\max}(X_k) \leq R \quad \text{almost surely.}$$

*Define*

$$Y := \sum_{k} X_k \quad \text{and} \quad \mu_{\min} := \lambda_{\min}(\mathbb{E}[Y]) = \lambda_{\min}\left(\sum_{k} \mathbb{E}[X_k]\right).$$

*Then, for any $\delta \in (0, 1)$,*

$$\Pr\left[\lambda_{\min}\left(\sum_{k} X_k\right) \leq (1-\delta)\mu_{\min}\right] \leq d\left[\frac{e^{-\delta}}{(1-\delta)^{1-\delta}}\right]^{\mu_{\min}/R}.$$

**Lemma 7** (Determinant-trace inequality, Lemma 10 in Abbasi-Yadkori et al. [2]). *Suppose $X_1, X_2, \ldots, X_t \in \mathbb{R}^d$ and for any $1 \leq s \leq t$, $\|X_s\|_2 \leq L$. Let $\overline{V}_t = \lambda I + \sum_{s=1}^{t} X_s X_s^\mathsf{T}$ for some $\lambda > 0$. Then,*

$$\det(\overline{V}_t) \leq \left(\lambda + \frac{tL^2}{d}\right)^d.$$

**Lemma 8** (Azuma-Hoeffding inequality). *If $\{X_t\}_{t \geq 0}$ is a supermartingale corresponding to a filtration $\{\mathcal{F}_t\}_{t \geq 0}$, and $\{X_t\}_{t \geq 0}$ satisfies $|X_t - X_{t-1}| \leq c_t$ for all $t = 1, \ldots, T$. Then for any $\varepsilon > 0$, we have*

$$\Pr[X_T - X_0 \geq \varepsilon] \leq \exp\left(-\frac{\varepsilon^2}{2\sum_{t=1}^{T} c_t^2}\right).$$

