# OpenReview forum: "Towards Efficient Conversational Recommendations: Expected Value of Information Meets Bandit Learning"
_ACM.org/TheWebConf/2025/Conference — WWW 2025 Poster_

### Official Review · Reviewer_xxrL · 2024-11-26

**Novelty:** 3
**Technical Quality:** 4

**Review:**

This paper proposes to enhance conversational recommender systems by integrating the Expected Value of Information (EVOI) with bandit learning. The authors introduce two key modules: gradient-based EVOI and smoothed key term contexts, which results in two new algorithms, ConTS-EVOI and ConUCB-EVOI. Experiments seem to demonstrate the effectiveness of the proposed method.

**Strength:**

+ The paper is generally well-written
+ The studied problem is very important and the two modules make sense.
+ Experiments seem to show the effectiveness of ConTS-EVOI and ConUCB-EVOI compared with baselines.

**Weakness:**

- I think the main issue of this paper is that, while it is set in the context of conversational recommender systems (CRS), the proposed improvements seem to be independent of the CRS setting. The gradient-based EVOI introduced in Section 4.1 is not specifically designed for CRS. While the smoothed key-term context can be connected with CRS, the proposed method in fact adds Gaussian noise to the feature vector, which can be generalized to any bandit algorithm. If the authors aim to improve the bandit algorithm, more settings should be considered. If the contribution is to conversational bandit for recommendations, it is crucial to justify how each module tightly couples with the CRS setting.

**Questions:**

Please refer to the weakness in the main review

**Reviewer Confidence:**

3: The reviewer is confident but not certain that the evaluation is correct

**Scope:**

4: The work is relevant to the Web and to the track, and is of broad interest to the community

---

### Official Review · Reviewer_o7fJ · 2024-11-27

**Novelty:** 3
**Technical Quality:** 4

**Review:**

This paper introduces a novel framework for conversational recommender systems by integrating the Expected Value of Information (EVOI) into a bandit learning model. The motivation is claimed to address the limitation of previous bandit learning approaches that their query selection strategies yield only marginal regret improvements over non-conversational approaches and computational effectiveness issues. The proposed ConTS-EVOI and ConUCB-EVOI optimize query selection using a gradient-based EVOI and smoothed key term contexts. The experimental results demonstrate the effectiveness of their approach. The pros and cons of this paper are concluded as follows.

Strengths:
1.This paper proposes an efficient conversational recommender system framework by integrating of gradient-based EVOI with conversational bandits, as it addresses both the computational challenges of Bayesian EVOI and the exploration limitations in conversational bandits, enhancing the efficiency and accuracy of recommendations.

2. The authors provide a rigorous theoretical analysis showing that their proposed algorithms achieve tighter regret bounds, offering improvements over standard conversational bandit methods.

3.The paper evaluates the proposed algorithms on both synthetic and real-world datasets, demonstrating its effectiveness.

**Questions:**

Weakness:
1. The motivation is not clear. For example, the paper claims that conventional EVOI's Bayesian updating is computationally intensive. However, it lacks evidence or real-world data analysis to show whether such limitation leads to bad consequences. Put it another way, do all previous works perform poorly due to this limitation?

2. This paper is also placed poorly. In discussing the drawbacks of current conversational bandit approaches, the paper primarily focuses on the isolated limitations of EVOI and traditional bandit methods. It lacks a systemic analysis of the overarching issues in current bandit methods. This narrow framing makes it difficult to assess the method's contribution to theory and application. Without this, the contribution of this paper is incremental.

 3. if the proposed method can be applied to real-world industry cases, given that the Bandit algorithm can have scalability and robustness issues in large-scale and interactive recommender systems. would be beneficial to discuss these concerns and potential adaptations to existing SOTA recommendation models.

**Reviewer Confidence:**

3: The reviewer is confident but not certain that the evaluation is correct

**Scope:**

4: The work is relevant to the Web and to the track, and is of broad interest to the community

---

### Official Review · Reviewer_eaqG · 2024-11-29

**Novelty:** 5
**Technical Quality:** 5

**Review:**

This paper focuses on the exploration of query selection in conversational recommender systems. Previous work has measured and optimized the quality of queries through research on Expected Value of Information (EVOI) and Conversational Bandits. However, computational cost, myopia, and marginal improvement issues have hindered further progress in this area. This work integrates EVOI into Conversational Bandits to complement each other. EVOI provides effective query selection strategies, while Conversational Bandits offer theoretical guarantees for long-term performance.

1. The integration of the two existing research routes makes full use of their respective strengths.

2. The arguments presented in this paper have been validated both theoretically and experimentally, ensuring the soundness of the paper.

3. This paper is well-written and logically clear.

Weaknesses and Concerns:

1. The gradient-based EVOI proposed in this paper is essentially an approximation of the EVOI concept, and many such works have been conducted in the past. However, it seems that this paper does not provide a theoretical or experimental comparison between the proposed method and previous approximation-based approaches. In particular, regarding computational resource consumption, does the proposed method offer any advantages?

**Questions:**

See in Weaknesses and Concerns

**Reviewer Confidence:**

4: The reviewer is certain that the evaluation is correct and very familiar with the relevant literature

**Scope:**

4: The work is relevant to the Web and to the track, and is of broad interest to the community

---

### Official Review · Reviewer_En3j · 2024-12-02

**Novelty:** 6
**Technical Quality:** 5

**Review:**

This paper presents a comprehensive analysis of integrating EVOI with conversational bandit frameworks, addressing computational complexity while maintaining robust theoretical foundations. The authors propose two novel mechanisms: a gradient-based EVOI method for efficient query selection and a randomized query perturbation technique to enhance exploration. These mechanisms are integrated into two tailored algorithms, ConTS-EVOI and ConUCB-EVOI, which build on established methods like conversational Thompson Sampling and LinUCB. The strong theoretical validation, including improved regret bounds, ensures the credibility of the approach, while the focus on both exploratory query selection and long-term regret minimization makes it versatile for various recommendation systems. Also, authors have done the evaluation on variety of datasets particulary for regret analysis.

Along with the above strengths, I would like to highlight the following points to be improved. While the paper's use of a stochastic gradient-based approach helps reduce computational complexity, it heavily depends on the choice of hyperparameters, such as the learning rate $\alpha$, which the authors set to 0.1. However, the reasoning behind this choice is unclear, and the effects of changing this value are not explored. The idea of using smoothed key term context adds some novelty, but its impact seems limited and highly dependent on the specific application or user preferences. Randomly changing queries could also risk introducing irrelevant ones, which the authors should discuss in detail. Although computational efficiency is highlighted as a key strength, the results section does not provide a clear evaluation of this aspect. Finally, the methods assume a linear reward structure, which might not align with many real-world scenarios, potentially limiting how widely the approach can be applied.

**Questions:**

Please refer to the second paragraph in the review. Addressing questions and concerns raised will add more value to the paper. Adding a more discussion based on those facts will be sufficient.

**Reviewer Confidence:**

2: The reviewer is willing to defend the evaluation, but it is likely that the reviewer did not understand parts of the paper

**Scope:**

3: The work is somewhat relevant to the Web and to the track, and is of narrow interest to a sub-community

---

### Official Review · Reviewer_ARYp · 2024-12-03

**Novelty:** 5
**Technical Quality:** 5

**Review:**

The authors propose a new gradient based EVOI that removes the need to do complex Bayesian updates, and also introduce smoothed key term contexts that add some noise and random perturbation to uncover and explore very specific user preferences. The authors prove and show that their methods achieve substantially tighter regret bounds.

The paper is well written and motivated and the authors use the linearity in the formulation of conversational bandits to come up with a stochastic gradient descent based parameter estimation for updating both the preference vectors, PER and key term. The authors further show that the cumulative regrets of both ConTS-EVOI and ConUCB-EVOI scale in $O(d\sqrt(Tlog(T) + C)$, and the authors provide a lot of experiments justifying their results.

In terms of technical contribution, my feeling is that the gradient based EVOI seems like a fairly obvious thing to try given the very linear structure of the problem and I am unsure about the novelty of the paper, and the smoothing of key terms also seems fairly standard. However, the theoretical guarantees they show are very interesting, and I am leaning towards an acceptance of this work.

**Questions:**

1) For the smoothed key term context, how do you make sure that the term doesn't massively deviate from its original meaning? One of the reasons to using key terms is to divide up information properly - how can we be sure that with this smoothing term, we still have such guarantees?

2) In terms of ConvRec evaluations, I feel this paper is still lacking in evaluations for datasets that have natural conversational needs and I urge the authors to test on more relevant datasets.

**Reviewer Confidence:**

2: The reviewer is willing to defend the evaluation, but it is likely that the reviewer did not understand parts of the paper

**Scope:**

4: The work is relevant to the Web and to the track, and is of broad interest to the community